# *In Silico* Simulations Reveal Molecular Mechanism of Uranyl Ion Toxicity towards DNA-Binding Domain of PARP-1 Protein

**DOI:** 10.3390/biom13081269

**Published:** 2023-08-20

**Authors:** Egor S. Bulavko, Marina A. Pak, Dmitry N. Ivankov

**Affiliations:** Center for Molecular and Cellular Biology, Skolkovo Institute of Science and Technology, Bolshoy Boulevard 30/1, Moscow 121205, Russia

**Keywords:** metalloproteins, PARP-1, DNA-binding domain, zinc finger, uranium, uranyl ion, reaction mechanism, free energy profiles, QM/MM, QM/MM molecular dynamics

## Abstract

The molecular toxicity of the uranyl ion (UO_2_^2+^) in living cells is primarily determined by its high affinity to both native and potential metal-binding sites that commonly occur in the structure of biomolecules. Recent advances in computational and experimental research have shed light on the structural properties and functional impacts of uranyl binding to proteins, organic ligands, nucleic acids, and their complexes. In the present work, we report the results of the computational investigation of the uranyl-mediated loss of DNA-binding activity of PARP-1, a eukaryotic enzyme that participates in DNA repair, cell differentiation, and the induction of inflammation. The latest experimental studies have shown that the uranyl ion directly interacts with its DNA-binding subdomains, zinc fingers Zn1 and Zn2, and alters their tertiary structure. Here, we propose an atomistic mechanism underlying this process and compute the free energy change along the suggested pathway. Our Quantum Mechanics/Molecular Mechanics (QM/MM) simulations of the Zn2-UO_2_^2+^ complex indicate that the uranyl ion replaces zinc in its native binding site. However, the resulting state is destroyed due to the spontaneous internal hydrolysis of the U-Cys162 coordination bond. Despite the enthalpy of hydrolysis being +2.8 kcal/mol, the overall reaction free energy change is −0.6 kcal/mol, which is attributed to the loss of domain’s native tertiary structure originally maintained by a zinc ion. The subsequent reorganization of the binding site includes the association of the uranyl ion with the Glu190/Asp191 acidic cluster and significant perturbations in the domain’s tertiary structure driven by a further decrease in the free energy by 6.8 kcal/mol. The disruption of the DNA-binding interface revealed in our study is consistent with previous experimental findings and explains the loss of PARP-like zinc fingers’ affinity for nucleic acids.

## 1. Introduction

Uranium (U) is a poisonous heavy metal that is primarily of concern due to inherent radioactivity. However, under natural conditions, its chemical toxicity can pose an even greater risk [1]. The uranyl ion (UO_2_^2+^), the most stable soluble form of uranium, has shown the ability to interfere with numerous biochemical processes [2]. For instance, uranyl can substitute iron in the natural metal-binding site of hemoglobin, significantly reducing the amount of its oxygenated form in the blood [3]. Uranyl also disrupts the formation of the *Cyt c*/*Cyt b_5_* complex, which is involved in cell respiration [4].

Several experimental and computational studies have addressed the ability of the uranyl ion to induce conformational changes in individual proteins and disrupt their complexes [5,6]. However, the structural mechanisms underlying the uranyl-mediated disruption of DNA-protein interactions remain understudied. It has long been known that uranium poisoning is associated with DNA repair deficiency [7,8]. In 2016, Cooper et al. experimentally showed that the uranyl ion is a direct inhibitor of Poly [ADP-ribose] polymerase 1 (PARP-1), a eukaryotic enzyme that plays a crucial role in recognizing and repairing DNA breaks [9] (Figure 1). The authors hypothesized that the zinc finger domains Zn1 and Zn2, which contribute significantly to DNA damage recognition, are the main targets of uranium. This hypothesis was partially confirmed by Zhou et al., who conducted experiments with an isolated fragment of the Zn1 domain containing the zinc-binding site [10]. They found that the uranyl ion disrupts the native tertiary structure of the fragment upon binding, but it does not show thermodynamic preference over zinc, which seems contradictory to the results achieved by Cooper et al. [9].

In this study, we employed a wide range of computational biochemistry techniques to investigate the structural properties of the interaction between the uranyl ion and the PARP-1 zinc finger Zn2, aiming to elucidate a possible molecular pathway leading to the decreased affinity of the domain to nucleic acids. We found that upon replacing the zinc ion in its native binding site with uranyl, the coordination environment changed from tetrahedral to planar, resulting in the leaving of His159 from the coordination environment. Furthermore, we discovered that the only remaining coordination bond between the N-loop/UO_2_^2+^ subcomplex and the core residue Cys162 undergoes spontaneous internal hydrolysis with a positive enthalpy change of +2.8 kcal/mol. However, due to the increased conformational entropy of the relaxed loop, the overall reaction free energy change was found to be −0.6 kcal/mol. Finally, we identified a subsequent reorganization of the binding site, facilitated by the coordination of the uranyl ion to the Glu190/Asp191 acidic cluster, resulting in a decrease in free energy by 6.8 kcal/mol. These significant perturbations in the tertiary structure of the domain provide a satisfactory explanation for the toxic effects of the uranyl ion and are consistent with observations published earlier [9,10], thus, resolving the apparent contradiction between the previous experimental results.

## 2. Models and Methods

We extracted the X-ray crystalline structure of the Zn2 zinc finger of human PARP-1 (PDB ID 3ODC) [12] as the starting point for our study. The MIB2 web server (http://bioinfo.cmu.edu.tw/MIB2, accessed on 13 October 2022) [13] was utilized to identify potential metal-binding sites within the zinc finger structure. We used several types of doubly charged metal ions offered by the program (Zn^2+^, Co^2+^, Pt^2+^, Hg^2+^, Ni^2+^) as a template for our binding site search.

For molecular dynamics simulations, we dissolved the protein in a rectangular 7.1 × 6.8 × 6.3 nm^3^ water box and added sodium and chloride ions to neutralize the system and make the NaCl concentration equal to 150 mM. We utilized the CHARMM36 force field [14,15,16] and TIP3P water model [17] to generate topology that was then converted from CHARMM to GROMACS format. All actions described above were performed in VMD workspace (version 1.9.4a53) [18] using built-in psfgen and topogromacs [19] plugins for building and manipulating topology.

We performed classical molecular dynamics simulations by employing the GROMACS 2021.6 program package [20]. Typical simulation setup implied utilization of a V-rescale thermostat [21] (T = 300 K, time constant 1 ps) with separated temperature coupling for solute and solvent molecules and isotropic Parrinello–Rahman barostat (P = 1 atm, time constant 5 ps, compressibility 1 atm^−1^). Covalent bonds that contain hydrogen atoms were constrained with LINCS [22] and SHAKE [23] algorithms for solute and water respectively. We set Coulomb and Van der Waals (VdW) cutoffs to 1.2 nm. Long-range electrostatics were computed using the Particle Mesh Ewald (PME) method [24], and the overall integration time step was set to 2 fs.

To study the electronic structure and bonding properties of complexes with uranyl and zinc ions we performed pure quantum chemistry calculations. More than 10 four- and five-coordinated complexes varying in coordination sphere composition were explored. The set of possible ligands included hydroxyl, acetate (CH_3_COO^−^), and methanethiolate (CH_3_S^−^) ions, as well as water molecules. We conducted geometry optimizations at the DFT/PBE0/D3BJ/ZORA level of theory [25,26,27], which was proven to be an adequate choice for calculations including uranium compounds [28,29]. Pople-style basis set 6-311++G** [30,31] was utilized for light atoms and zinc, while SARC-ZORA-TZVP [32] was used for uranium. Additionally, we applied the implicit CPCM solvation model [33]. We performed these calculations in ORCA (version 5.0.1) program package [34]. We processed final wave functions using Multiwfn (version 3.8) program [35].

The chemical transformations occurring in the proteins’ metal-binding sites and corresponding free energy changes were studied by QM/MM molecular dynamics in combination with bias potentials (umbrella sampling, US) [36]. The quantum subsystem contained the uranyl ion, surrounding amino acids’ side chains (namely, Cys125, Cys128, His159, and Cys162 at the first stage and Cys125, Cys128, and Glu190 at the second), along with two catalytic water molecules. In corresponding figures below, quantum partitions (excluding alpha carbons) are drawn opaque. Forces acting in the QM part were described at DFT/PBE0/D3BJ level of theory with 6-31G** basis for light atoms and LANL2DZ [37] for uranium. The metal’s non-valence electrons were replaced by HayWadt effective core potential (ECP) [38] for further reduction of computational load. QM-MM interactions were dealt with in terms of electrostatic embedding; Mulliken atomic charges were computed for the quantum subsystem at every step of molecular dynamics. We used the link atom scheme to resolve the boundary effects. We described the classical subsystem with the CHARMM36 force field, and the solvent—with TIP3P water model. We performed QM/MM calculations using NAMD (version 2.14)/ORCA program interface [39]. Hybrid molecular dynamics setup implied usage of Langevin thermostat (T = 300 K, damping coefficient 5 ps^−1^), Nosé–Hoover Langevin piston barostat (P = 1 atm, oscillation period 200 fs, damping scale 100 fs), integration step 1 fs and the absence of any bond constraints. We set the non-bonded cutoff to 1.2 nm and computed long-range electrostatics with PME. The collective variables (colvars) module [40] was utilized to apply harmonic bias potential along the reaction coordinate in the US simulations (Appendix A). We used the weighted histograms analysis (WHAM) [41] and umbrella integration (UI) [42] methods to analyze the results of the US calculations.

We additionally evaluated the enthalpy of chemical reactions by comparing bare electronic energies of the quantum subsystem in initial and final states. From 5 ps simulations of reactant and product we randomly selected twenty snapshots and performed geometry optimizations of the quantum subsystem in the presence of point-charge correction conditioned by 1730 carefully selected solute and solvent atoms and absolute coordinate restraints applied to link hydrogens. We then compared the difference between average energies of optimal geometries to those resulting from the US simulations.

To parametrize uranium coordination complexes that correspond to the observed metastable states in terms of classical force field, we utilized a standard protocol implemented as the Force Field toolkit (ffTK) plugin [43] in VMD. Restrained electrostatic potential (RESP) partial charges were assigned to atoms; VdW parameters for uranyl were obtained from one of the previous studies [44].

We used the GIST (Grid Inhomogeneous Solvation Theory) methodology [45,46] implemented in the AMBER CPPTRAJ program [47] to estimate the solvation enthalpy and entropy of different protein states. A 5000 ns trajectory was computed for each state and clustered using the GROMACS cluster module (clustering method—gromos, cutoff—0.5 nm). The most representative structure from each cluster underwent additional 500 ns simulations (resulted in 50,000 snapshots in total) with protein atoms positions restrained. These trajectories were analyzed using GIST. The ParmEd python library [48] was utilized to convert topology files to a format suitable for CPPTRAJ. The GIST grid size and center were chosen to include the whole protein and a 10 Å solvent buffer zone. Grid spacing was set to 0.5 Å for water entropy and 1.0 Å for water energy calculations. Multiple GIST runs were performed, gradually increasing the frames analyzed until the target thermodynamic properties values reached a plateau. Water–water energy was calculated from a pairwise interaction matrix [46]. The total solvation free energy included translational and orientational water entropy terms, water–solute and water–water energy. For protein states represented by multiple structures, we calculated solvation energy by weighted averaging.

We estimated configurational enthalpy in a particular state as a sum of internal electrostatic and VdW interactions. We omitted all bonded terms because they are commonly relaxed, on average. We took a hundred random snapshots from each cluster and processed them using the g_mmpbsa program [49] to evaluate the target properties. We utilized the PARENT program [50] to calculate configurational entropy by MIST (Mutual Information Spanning Tree) method. We analyzed 1500 ns trajectories (with coordinates written every ps). The number of bins was set to 50 for 1D histograms and to 2500 for 2D ones.

## 3. Results and Discussion

### 3.1. Structural Properties of Uranyl-Protein Complex

We began our investigation by conducting a structure-based search for metal-binding sites on the surface of the Zn2 zinc finger. Apart from the native zinc-binding site, we identified two additional potential sites (Figure 2A). Both consist of neighboring acidic residues, specifically Asp and Glu, which form a cluster capable of accommodating up to four coordination covalent bonds (CCBs). One of these potential sites is located at the terminus of the base-stacking loop, while the other resides within one of the alpha helices in the core region. Considering the previously observed zinc loss upon exposure of full-length PARP-1 and its zinc finger to uranyl salts [9,10], we focused our further investigation on the native binding site.

The native zinc-binding site involves four amino acid side chains coordinating the zinc ion. Specifically, the N-loop contains two cysteine residues, which, along with a histidine and a cysteine from the protein’s core, form a tetrahedral environment for the zinc ion (Figure 1). We performed quantum chemistry calculations (for details see “Models and Methods”) and found that the zinc–sulfur bond in the complex has a characteristic length of 2.3 Å, and the zinc-nitrogen bond is 1.8 Å long. The average bond orders for Zn–S and Zn–N bonds were determined to be 0.84 and 0.91, respectively. Additionally, the zinc charge was found to decrease from +2.0 to +0.9 upon binding to the protein. These findings indicate a predominant covalent contribution in the binding process, signifying the formation of a strong and stable complex. As a result, the N-loop becomes tightly bound to the protein’s core, which is crucial for achieving a high affinity to DNA [12].

The uranyl ion is known to prefer a flat coordination sphere with four, five, or six ligands. Additionally, due to its strong acceptor properties, it exhibits a higher affinity to charged ligands compared to neutral ones. Consequently, we anticipated significant perturbations upon substituting zinc with uranium in the native binding site. Our QM/MM molecular dynamics simulations revealed that His159 dissociates from the coordination environment, and the remaining cysteine residues, along with two water molecules, formed an irregular pentagon lying in a plane perpendicular to the O-U-O axis (Figure 2B). The bonds between uranium and sulfur atoms in the complex were longer and weaker compared to the corresponding zinc–sulfur bonds, with the average length of 2.75 Å and the bond order of 0.65. The distances between uranium and oxygen atoms in the coordinated water molecules were comparable to the uranium-sulfur distances, approximately 2.71 Å. However, the corresponding bond orders were relatively small, around 0.28. Moreover, only a small charge of −0.13 was transferred from water to uranium, indicating a lower covalent contribution to the U–H_2_O bond.

### 3.2. Coordination Environment Determines the Complex Stability

The precise positioning of the N-loop is crucial for DNA recognition, and zinc plays a critical role in maintaining its proper orientation within the protein structure [12]. Therefore, we aimed to investigate the extent to which uranyl ion holds the connection between the N-loop and the protein’s core.

We analyzed how the composition of the coordination environment affects the overall stability of the complex. We examined over 10 complexes and evaluated four bond properties: bond length, Mayer order, IBSI (Intrinsic Bond Strength Index), and transferred Mulliken charge. Mayer bond order has been proven to be a valuable metric for assessing complexation effectiveness [51], and thus, we focused primarily on its correlation with the content of the coordination sphere.

The results of our calculations are presented in Appendix A, and Figure 3 highlights two important findings. Firstly, as hydroxyl anions (OH^−^) are introduced into the coordination sphere, there is a gradual decrease in the average bond order between uranium and sulfur atoms. This suggests that the presence of small charged ligands weakens the bond strength between uranium and sulfur. Secondly, as a general trend, the uranium–sulfur bond order in five-coordinated complexes is lower compared to four-coordinated complexes, indicating the significant role of steric factors in determining the stability of these complexes. For example, if the coordination sphere consists of two CH_3_S^−^ ions and two water molecules, the bond order is 0.85. However, it drops to 0.36 when the environment includes three CH_3_S- and two OH-groups. Interestingly, the uranium–water and uranium–hydroxyl interaction properties remain unchanged when varying the complex composition. The deviations in bond lengths and orders from one complex to another do not exceed 11% (Appendix A). These findings demonstrate that under specific conditions, the uranium–sulfur bonds can weaken and potentially dissociate.

### 3.3. Cysteine 162 Decomplexation Leads to the Loss of Zinc Finger Tertiary Structure

The U–Cys162 bond is of particular importance in our study, as its disruption could result in the loss of connectivity between the zinc finger domains (Figure 2B) and potentially lead to a change in orientation of the N-loop, which is undesirable for DNA recognition [12]. As demonstrated earlier, the initial coordination environment composition includes three cysteines, two water molecules, and no hydroxyl ions. This configuration corresponds to a relatively high bond order of 0.78. Replacing one water molecule with hydroxyl anion would yield a complex with the formula [UO_2_(SCH_3_)_3_(OH)(H_2_O)]^2−^. Despite quantum chemistry calculations indicated its instability, we estimated the uranium–sulfur bond order for this modified complex to be lower than 0.49. The process of water deprotonation could occur concurrently with cysteine dissociation within a process called internal hydrolysis:
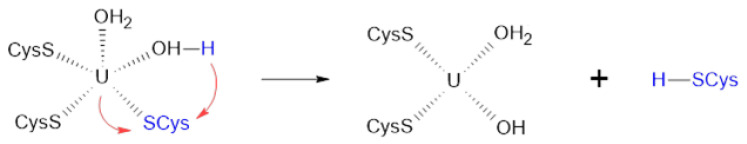


It implies the transfer of a proton from water to cysteine, which further facilitates the breaking of the uranium–sulfur bond, since the uranyl ion exhibits stronger acidic properties than hydrogen sulfide and its derivatives [52].

To evaluate the energy of chemical reactions, the umbrella sampling (US) method is commonly used. It takes into account both the enthalpy and entropy contributions. However, it is not possible to separate these energy terms. Therefore, to estimate the hydrolysis enthalpy alone, we employed the protocol described in the Section 2.

The free energy profile of the hydrolysis reaction obtained from the US calculations is presented in the Appendix A. It is evident that the unhydrolyzed state (state I, SI) exhibits a lower free energy by 2.8 kcal/mol compared to the hydrolyzed state (state II, SII), indicating its higher chemical stability. The energy barrier separating the metastable states is 9.1 kcal/mol, which refers to the expected transition time about 10–100 ns. As a result, the hydrolysis reaction does not meet kinetic limitations while moving to equilibrium.

The enthalpy of hydrolysis was found to be +2.8 ± 0.2 kcal/mol, which aligns perfectly with the value obtained from the US calculations. This suggests that the entropy change upon the reaction is close to zero, and the enthalpy is the primary driving force in this process.

However, considering the chemical reaction alone is insufficient to provide an adequate representation of the existing equilibrium. The decomplexation of Cys162 causes the N-loop to disconnect from the protein’s core, potentially increasing the mobility of this subdomain. As a result, there could be an increase in configurational entropy, leading to a decrease in the relative free energy of SII. On the other hand, solvation entropy reduction due to the growth of the solvent-accessible area upon partial unfolding typically interferes with the configurational effect [53]. Thus, considering solvation and conformation thermodynamics is crucial for obtaining comprehensive insight into the relative stabilities of the corresponding states.

Classical molecular dynamics simulations revealed that the lifetime of the hydrolyzed state in the native configuration does not exceed 20–30 ns (Figure 4). The system predominantly exists in two conformations, with a ratio close to 4:1. In one conformation, the N-loop is fully exposed to the solvent and exhibits high-amplitude oscillations. In the other conformation, the terminus of the loop is closer to the protein’s core, and oscillations are predominantly absent. It is worth noting that the apo-protein, which is not bound to any metal, maintains its original structure even during 3000 ns simulations, as indicated by the grey line on the RMSD plot (Figure 4).

The total free energy difference between SII and SI has the following expression:$$∆G_{tot}^{I\to II}=∆H_{reac}^{I\to II}+\left(∆H_{conf}^{I\to II}-T∆S_{conf}^{I\to II}\right)+\left(∆H_{solv}^{I\to II}-T∆S_{solv}^{I\to II}\right).$$

Here, $∆H_{reac}^{I\to II}$ stands for hydrolysis enthalpy, which we showed to be equal to the reaction free energy. In the first parentheses, configurational terms are written, namely, differences of intraprotein enthalpy and conformational entropy. The terms from the second parentheses account for solvation thermodynamics. Finally, $∆H_{solv}^{I\to II}$ describes how water-water and water-solute interactions enthalpies change upon transition, and $T∆S_{solv}^{I\to II}$ shows the difference of water translational and rotational entropies between SII and SI. A detailed description of corresponding computational protocols is presented in the Section 2.

Appendix A provides the final computed values of $∆G_{tot}^{I\to II}$ components. The sum of configurational and solvent enthalpies is close to zero, which aligns with the fact that protein folding is primarily driven by entropic factors [54]. Excluding the hydrolysis energy, the free energy difference between states II and I is −3.4 kcal/mol. This indicates that in the equilibrium mixture, the fraction of SI will be less than 0.5%. As mentioned earlier, once destroyed within a few dozen nanoseconds, the initial state does not regenerate, even within 10^3^ ns-long simulations. Considering that only the native zinc finger’s fold is capable of binding nucleic acids, this would inevitably result in domain malfunction. However, it should be noted that the unfavorable nature of hydrolysis increases the final free energy difference to −0.6 kcal/mol, which may not be sufficient to completely disrupt the functionality of the Zn2 domain. Firstly, upon accounting for the partial inaccuracies of the computational methods, even the sign of the free energy difference could change. Secondly, an equilibrium will be established where comparable fractions of both states coexist. Therefore, we sought to investigate additional structural and chemical transformations that occur after hydrolysis and govern the stabilization of the protein’s “broken” (non-functional) state.

### 3.4. Binding Site Reorganization

We hypothesized that the association of the hydrolyzed uranyl complex with one of the potential metal-binding surfaces found (Figure 2A) might firm a non-functional state. Among the two potential binding sites investigated, only those composed of Glu190 and Asp191 residues demonstrated the ability to interact with the N-loop. Even without explicit modeling of the CCB between uranium and the acidic cluster, the corresponding conformation of the loop (Figure 5A) remained intact during the 10^3^ ns-long molecular dynamics simulations.

Furthermore, our molecular dynamics simulations indicated that only Glu190 had an optimal orientation to coordinate the uranyl ion. To propose a potential reaction mechanism, we assessed the stability of uranyl complexes involving two methylthiolate groups, one acetate ion, and varying numbers of water molecules and hydroxyl anions. We found that five-coordinated complexes, as well as those with carboxylate as a bidentate ligand, were energetically and structurally unfavorable. Specifically, the uranium–sulfur bond order in such complexes was less than 0.1, indicating the absence of coordination bonding. Therefore, it was most likely that the association would proceed through the substitution of one of the existing ligands. Given that the uranium–water CCB exhibited the lowest bond order, the water molecule is likely to be substituted:
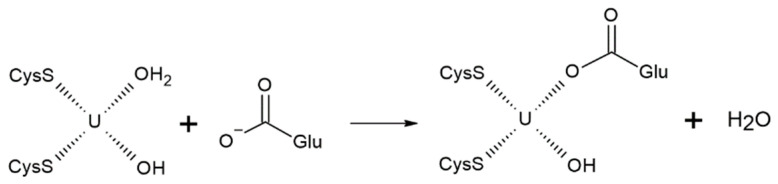


Moreover, we cannot determine with certainty whether the hydroxyl anion formed in the coordination sphere upon hydrolysis remains deprotonated. However, the association with the carboxylate group of the hydroxyl-containing complex [UO_2_(SCH_3_)_2_(OH)(H_2_O)]^−^ would be less favorable compared to [UO_2_(SCH_3_)_2_(H_2_O)_2_]. Therefore, the energy of the reaction involving the former complex would provide an upper estimate.

According to our calculations, in the final state (Figure 5B), the bond between uranium and sulfur has a length of 2.77 Å and a bond order of 0.51. In turn, the CCB between uranium and the carboxyl oxygen of Glu190 is relatively short (2.31 Å), but its bond order indicates that it is not very strong, with a value of 0.59. However, if we assume that the OH-group is going to be reprotonated, the U–SCys bond order will increase to 0.68, and U–OGlu—to 0.88. Therefore, from a chemical perspective, the final complex (SIII) should be more stable than the initial one (SI).

Similar to previous calculations, we computed both the free energy change along the association reaction pathway (Appendix A) and the corresponding enthalpy. This time, the energy of the product (SIII) was significantly lower than that of the reactant (SI). Notably, the magnitude of the energy difference exceeds that of the internal hydrolysis, indicating that state III is chemically more stable than any other state. The activation barrier for the forward reaction turned out to be 1.5 kcal/mol, which makes reaching the equilibrium extremely fast. However, in contrast to the internal hydrolysis reaction, the association free energy (−12.1 kcal/mol) was slightly lower than its enthalpy (−11.7 ± 0.3 kcal/mol). This suggests a non-zero contribution from the entropic term. The water molecule released during the reaction gains the mobility characteristic of bulk solvent. Considering that the QM/MM molecular dynamics simulation covers a short time period and the US setup imposes restrictions on the translational movements of perturbed molecules, we assume that the 0.4 kcal/mol difference arises from the increased rotational entropy of water. Bulk TIP3P water has translational ($TS_{trans}^{TIP3P}$) and rotational ($TS_{rot}^{TIP3P}$) entropies of 4.2 and 1.0 kcal/mol, respectively (at T = 298 K) [55]. Hence, the increase in rotational mobility upon association was more than two-fold.

As for translational entropy, it is non-zero for coordinated water, which is still relatively loosely coupled to uranium (U–H_2_O bond order for SI and SII is 0.26 and 0.25, respectively) and sequentially being replaced by other solvent waters. However, since we cannot observe these replacements in the simulations, we are unable to estimate this term for our case. Therefore, we can only state that the increase in water translational entropy ($T{∆S}_{trans}^{I\to III}$) upon association is close to 4.2 kcal/mol.

For SI → SIII transition, the formula for total free energy difference is as follows:$$∆G_{tot}^{I\to III}=(∆H_{reac}^{I\to III}-T{∆S}_{trans}^{I\to III}-T{∆S}_{rot}^{I\to III})+\left(∆H_{conf}^{I\to III}-T∆S_{conf}^{I\to III}\right)+\left(∆H_{solv}^{I\to III}-T∆S_{solv}^{I\to III}\right)$$

These terms are either similar to those for the SI → SII transition or have been explicitly described. The computed values in Appendix A show that $∆G_{tot}^{I\to III}$ is expected to be close to −7.4 kcal/mol, indicating significant favorability of the “broken” state compared to the functional state. In state III, the orientation of the N-loop is far from optimal for effective DNA recognition, confirming that the protein’s DNA-binding interface is now destroyed. Thus, exposure of PARP-like zinc fingers to uranium salts results in irreversible tertiary structure transformations that impair their native functionality.

In our investigation of the chemical and conformational transformations occurring upon exposure of the PARP-1 zinc finger to the uranyl ion, we based our assumptions on the observed loss of DNA-binding activity and zinc depletion under such conditions. Importantly, our proposed mechanism aligns well with additional experimental evidence [9,10]. We have demonstrated that the complex formed between the zinc finger and uranyl undergoes significant alterations in its tertiary structure, while the apo-protein, lacking metal binding, maintains its native conformation over time [10]. Moreover, our mechanistic model reconciles the apparent discrepancy between the findings of Zhou et al., who observed a lack of thermodynamic advantage for uranyl over zinc in vitro [10] for the Zn1 domain fragment, and Cooper et al., who reported the replacement of zinc by uranyl in full-length PARP-1 in vivo [9]. Our study shows that the dissociation of the N-loop alone upon binding of the uranyl ion to the native metal-binding site does not provide significant thermodynamic favorability and can only lead to a partial decrease in the domain’s DNA-binding activity. Further, the uranyl ion forms only two CCBs with the zinc finger, which may explain the absence of its thermodynamic advantage over zinc observed by Zhou et al. [10]. The association of uranyl with an alternative metal-binding site, specifically the Glu190/Asp191 acidic cluster, which is present in the full-length zinc finger but not in the fragment used by Zhou et al. in their experiments, conditions the sufficient stabilization of non-functional state. The formation of a strong CCB between uranium and the carboxyl group of the glutamate residue reinforces the uranyl–protein interactions, potentially contributing to the thermodynamic advantage of uranyl over zinc.

## 4. Conclusions

By combining various computational biochemistry techniques, we have proposed a molecular model that elucidates the structural changes occurring in PARP-like zinc fingers upon their exposure to uranium salts. Our findings demonstrate that the binding of zinc and uranyl ions is competitive, with the uranyl ion occupying the metal-binding site that is partially composed of amino acid residues (Cys125, Cys128) responsible for zinc coordination in the natural functional conformational state. This is in line with experimental studies that have reported zinc loss upon interaction with uranyl ions [9,10]. However, we have shown that the uranyl ion complex in the native metal-binding site is unstable. The U–Cys162 coordination bond undergoes spontaneous hydrolysis, resulting in the disruption of the DNA-binding interface. The subsequent structural reorganizations facilitate the association of the uranyl ion with the carboxyl group of Glu190, leading to the formation of the final uranyl-binding site comprising residues Cys125, Cys128, Glu190, and one water molecule (or hydroxyl anion). The free energy difference between the “broken” and functional states of the protein-uranyl complex is approximately −7.4 kcal/mol, indicating a near-irreversible transformation. The significant perturbations in the tertiary structure of the domain elucidate the toxic effect of the uranyl ion and are consistent with observations published earlier, thus reconciling the apparent discrepancy between previous experimental results [9,10]. Our findings contribute to a better understanding of the toxic effects of uranium at the molecular level and can aid in future studies aimed at mitigating its detrimental impacts.

## Figures and Tables

**Figure 1 biomolecules-13-01269-f001:**
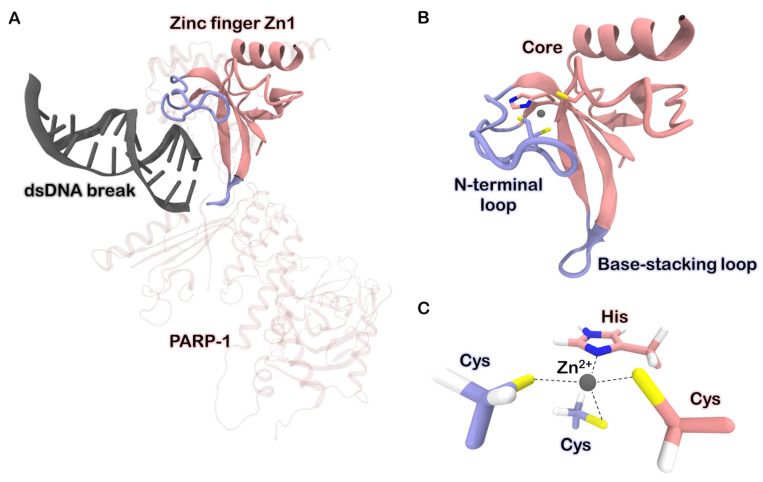
(**A**) Structure of PARP-1 protein bound to double-stranded DNA break by means of the zinc finger Zn1 (PDB ID 4DQY) [11]. The DNA-binding domain (DBD) of PARP-1 consists of three zinc finger motifs, Zn1, Zn2, and Zn3. The first two are homologous and make a major contribution to the recognition of DNA damages. (**B**) Structure of the zinc finger Zn1. The N-terminal loop (N-loop) recognizes the major groove of the DNA fragment, while the base-stacking loop interacts with nucleotides within the break. (**C**) Zinc ion binding site organization. Two cysteines located on the N-loop and one histidine and one cysteine protruding from the protein’s core form the tetrahedral coordination environment for metal.

**Figure 2 biomolecules-13-01269-f002:**
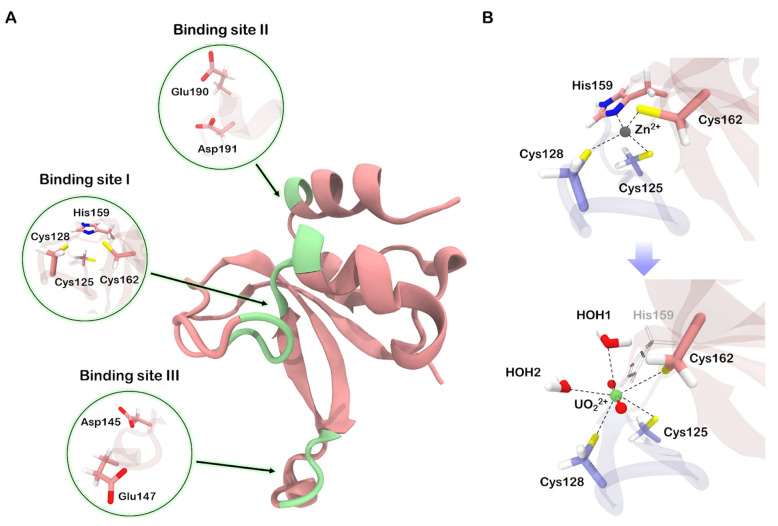
(**A**) Three metal-binding sites found on zinc finger surface using MIB2 server [13]. (**B**) Transformations occurring in the native binding site upon substitution of zinc by uranyl ion. HOH stands for water molecules. Quantum partitions (excluding alpha carbons) are drawn opaque.

**Figure 3 biomolecules-13-01269-f003:**
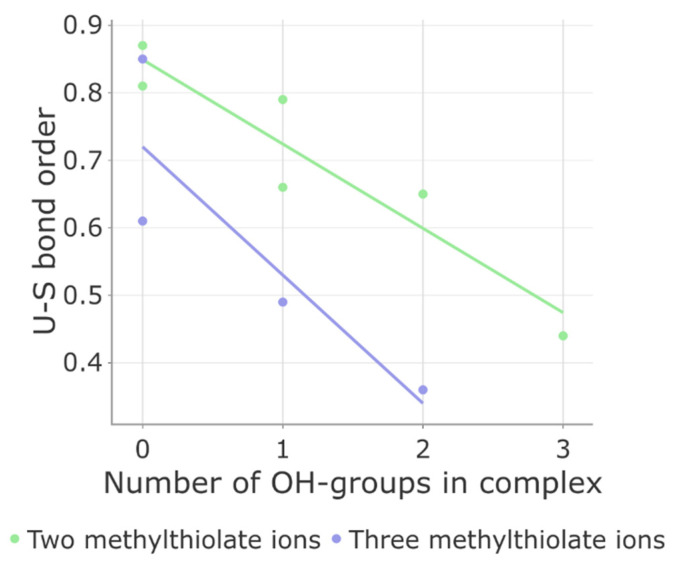
Correlation between Mayer bond order of uranium–sulfur (U–S) bond and ligand content, namely, the number of OH- and CH_3_S-groups, in uranyl ion complexes.

**Figure 4 biomolecules-13-01269-f004:**
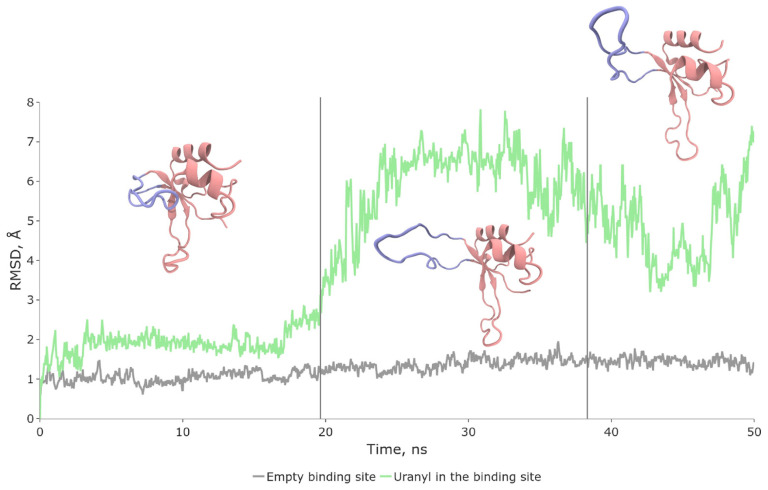
Conformational dynamics of hydrolyzed complex (SII, green color on RMSD plot) and apo-enzyme (grey). Vertical lines separate distinct configurations of SII. The most representative SII structures are shown.

**Figure 5 biomolecules-13-01269-f005:**
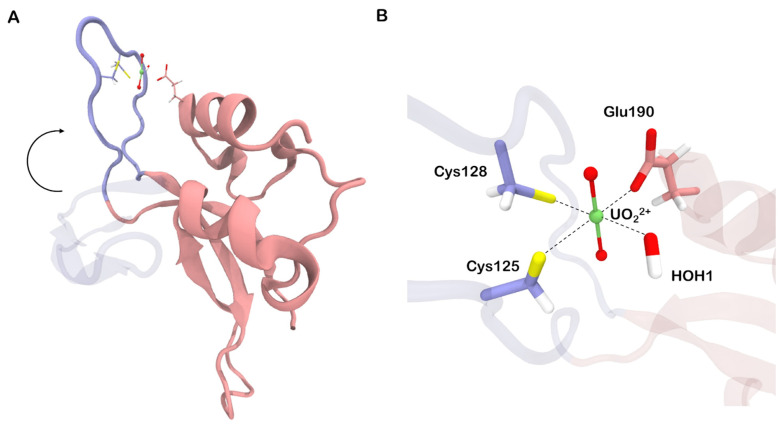
(**A**) Initial (violet, transparent) and final (violet, opaque) orientation of the N-loop. (**B**) Structure of uranyl-binding site in the final configuration. Quantum partitions (excluding alpha carbons) are drawn opaque.

## Data Availability

Data are contained within the article or Appendix A.

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
