# Peer review of "In Silico Simulations Reveal Molecular Mechanism of Uranyl Ion Toxicity towards DNA-Binding Domain of PARP-1 Protein"

_biomolecules, 2023, doi:10.3390/biom13081269_

Round 1
Reviewer 1 Report
This manuscript reports the multilayered computational study of the chemical and conformational effects of zinc replacement by uranyl ion in PARP-1 DNA-binding domain.
The narrative is well structured and organized. The reported artworks and tables, as well as the SI data, adequately support the reader in understanding the results and their interpretation.
The hypothesized chemical processes, i.e. histidine decomplexation, intramolecular hydrolysis of the U-S bond, and water(hydroxide)/carboxylate substitution, are conceivable and computational methodologies employed for the investigation of thermodynamics were well carried on.
In my opinion, the current version of the manuscript is suitable for pubblication in Biomolecules once the following minor issues were fixed:
- page 3, line 82: replace "took" with "extracted"
- page 3, line 90: replace "All actions described above we performed in VMD program" with "All actions described above were performed in the VMD workspace"
- page 3, line 93: replace "simulations in GROMACS" with "simulations by employing the GROMACS"
- page 3, line 104: replace "possible addends included" with "possible ligands included"
optional
- (page 6, line 233) and (page 7, line 263): replace "discoordination" with "decomplexation"
no comment
Reviewer 2 Report
The manuscript reports a multiscale computational study involving QM, classic MD and QM/MM enhanced sampling, to undrstand the effects of Uranyl anion poisoning on PARP1. In particular the authors have shown that the modification of the tertiary structure upon substitution of the native Zn by uranyl leads to the disruption of the structural motif responsible for the binding to DNA, thus impeding the activity of the protein.
Interestingly, the authors have also shown that while the substitution reaction itself is not providing enough thermodynamic driving force a subsequent complexation of acidic residues is leading to a strong stabilization and shifting of the equilibrium.
The manuscript is interesting and well written, and the authors have suceded in reconciling contrasting experimental evidences providing a clear picture of the effect of Uranyl anion on PARP. Thus, I believe this contribution is suitable for publication in Biomolecules pending some issues that the authors should comment on a revised version.
-In the abstract line 22 is quite ambigous and could be rephrased to avoid confusion between entropic effects, which are minimal, and global reorganization of the protein.
-h-bonds formulation (line 97) is ambigouos in LINCS/SHAKE the bonds involving hydrogen are constrained h-bonds may refer to hydrogen bonds which are a totally different physical phenomenon and which are, obviously, not constrained.
-A scheme or snapshots highlighting the quantum partition used in QM/MM would be useful to enhance the readability and better understand the strategy. In addition the authors should specify the collective variable used to perform the US simulation. By the way I suppose that this is performed using Colvar module, which should also be cited.
-Why has the most probable cluster structure been subjected to a further short MD simulation? The rationale behind this choice should be cited by the authors.
-The reason while the non-native binding sites identified by bio-informatic tools have been discarded should be discussed by the authors.
-The authors have noticed two water molecule entering the coordination sphere of U however those are, I suppose, not included in the QM partition could the author comment on this aspect and on the effects on the accuracy of their modelling?
-Instead of free energy cross-section profile I would use the term free energy profile or potential of mean force (PMF). Furthermore. I believe this results is of high importance and should be reported in the main text. More importantly, it appears that the activation energy for the hydrolysis reaction is small (~9kcal/mol) this aspect is highly relevant to confirm the feasaibility of the hypothesized pathway and should be reported and discussed in the main text. The same is true for the PMF involving the association of Glu to U. Indeed, in this case not only the driving force is high but the activation free energy is quite small of about 1 kcal/mol. This should also be pointed out since it is highly reinforcing the suitability of this mechanism.
Reviewer 3 Report
The most stable form of uranium under physiological conditions is uranyl ion, and the high toxicity of uranium might result from the ability of uranyl to interact with nucleotides and proteins, causing the disruption of their native functions. The studies of interaction of uranyl with the biomolecules found in human cells are of utmost interest since there is the vast experimental data showing the promiscuity of this radioactive dication in physiological milieu, for instance, its ability to bind to the blood proteins and proteins responsible for the bone growth. The article is well structured, well written, it contains the state-of-the-art computations and the obtained data are carefully analyzed, yielding the important conclusions on the mode of interaction of uranyl with the PARP1. The following issues should be addressed by the authors:
- Each of the cysteines in the studied Zn-finger may be either neutral or deprotonated, thus changing the charge of the Zn-finger as well as the pattern of electrostatic interactions in the system completely. The authors should study how the character of ZF's interaction with DNA changes when one or multiple cysteines are deprotonated. Indeed, such an additional study is indispensable since at pH 7.4 – typical in physiological conditions – the fraction of deprotonated Cys equals 5%.
- The authors should carefully describe how the MIB2 program was used. Which metal ions were used for this metal ion-binding site prediction program and why? There is no uranyl in the list of available probes in this program.
- The authors use the DFT functional PBE0-GD3BJ. Why? They should perform the benchmarking of most commonly used DFT functionals, and the corresponding table should be included into SI.
If these issues are properly addressed, this article can be published.
Round 2
Reviewer 3 Report
All my criticisms were properly addressed. To publish in the present form.